# On the Design of Aqueous Emulsions of Colophony Resin

**DOI:** 10.3390/polym15071691

**Published:** 2023-03-28

**Authors:** Isa B. D. Ingrez, Paula C. N. Ferreira, Davide Gameiro, Belmiro P. M. Duarte

**Affiliations:** 1Instituto Politécnico de Coimbra, Instituto Superior de Engenharia de Coimbra, Rua Pedro Nunes, Quinta da Nora, 3030-199 Coimbra, Portugal; 2Centro de Investigação em Engenharia dos Processos Químicos e dos Produtos da Floresta, Universidade de Coimbra, Rua Sílvio Lima, Pólo II, 3030-790 Coimbra, Portugal; 3Kemi Pine Rosins Portugal, S.A., Zona Industrial de Cantanhede, Lote 122, 3060-197 Cantanhede, Portugal

**Keywords:** product development, resin-in-water emulsions, colophony, experimental design, customer-centric design

## Abstract

Companies regularly face market pressure to develop products faster but they also need to simultaneously incorporate technological constraints, sustainability trends, and customer requirements into their designs, which requires the use of systematic procedures. Firms that exploit natural resources and convert them into high-value products are among them. However, the literature on the application of such systematic approaches to products of this type remains scarce, as they often requrire extensive experimental plans involving the testing and optimization of multiple formulations. Here, we propose a systematic approach to the design of pine-resin-in-water emulsions, which can be used to fabricate pressure-sensitive adhesives. The strategy is customer-centric in the sense that the customers’ specifications are integrated into the decision-making tool used to assess the quality of the formulations obtained through experiments. This tool uses *loss functions* to assess satisfaction with individual quality characteristics and multi-attribute decision-making methods to integrate them into an overall quality metric. Our framework is aligned with industrial practices and consists of three sequential stages: (i) screening of primary factors; (ii) optimization of secondary factors; and (iii) assessment of the experimental repeatability of the formulations. In each of these stages, the decision-making tool is used to “drive” the process of finding the optimal formulation.

## 1. Introduction

The chemical industry is constantly seeking opportunities to manufacture essential commodities and convert them into higher-value-added chemical-derived products [1]. The development of these chemicals requires (i) an economically viable process; (ii) a marketing strategy that aligns with market trends; and (iii) a sustainable production route and product (see, e.g., Harmsen et al. [2], Cussler and Moggridge [3]). Industries involved in the transformation of pine resin, an abundant natural and non-toxic raw material, into derivatives are currently facing the challenge to keep up with market demands. This industrial sector is often referred to as the *second industry* in the product value chain [4], where the *primary industry* is responsible for cleaning and separating the raw material into basic fractions (e.g., the separation of pine oleoresin into colophony and turpentine derivatives, see Zinkel and Russell [5]) and the *third industry* produces higher-value chemicals resulting from the transformation/incorporation of resin derivatives. Typically, the *second industry* produces derivatives in the form of elastomers, polymers for biomedical applications, coatings, adhesives, and surfactants [6], as well as food products and excipients for the fragrance and pharmaceutical industries [7]. Recently, colophony derivatives were recognized as exceptional and sustainable binders that can be used in combination with other materials to improve physical properties [8,9] or as repellent and anti-microbial coatings [10,11]. Additionally, their application to the fabrication of adhesives has been successfully investigated (see Hayashi et al. [12], Petrie [13], Unger et al. [14]).

Pressure-sensitive adhesives (PSAs) are a class of soft materials that adhere to different media with light pressure and short contact times, without a chemical reaction [15]. PSAs are often used in packaging applications (tapes and labels), medical applications, and baby and feminine hygiene applications, among others. Initially, PSAs were produced from natural rubber and they have predominantly been used in medical care. However, during the 1960s, rubber-based technologies were replaced by solvent-based synthetic polymers such as polyacrylates. Natural rubber emulsion-based technologies dominated the market until the late 1970s [16]. PSA formulations with improved thermal stability and different-molecular-weight versions of A-B-A block polymers were presented by Korpman [17].

Environmental concerns have forced industries to look for coating techniques other than solvent-based systems. In recent decades, effective cross-linked hot-melt adhesive systems have been developed to replace those based on natural rubber [18]. Water-based adhesive systems have become a viable alternative, and the increasing use of water emulsion-based systems ensures a greater selection of raw materials that can be incorporated into applications, meeting economic and sustainability requirements. Pine resin belongs to this group of materials and the industrial players in the market who have the ability to exploit it are encouraged to develop successful formulations.

PSAs are amorphous viscoelastic materials with a rheological behavior that is determined by the viscosity and elastic modulus. Both these properties, as well as the glass transition temperature (here represented by $T_{g}$), depend on the composition. The preparation of PSAs should ideally involve resin-in-water emulsions and the addition of plasticizers, diluents, emulsifiers, and stabilizers should be minimized [19]. In addition to the environmental impact of these agents, they can also have negative effects on the functional/structural properties of PSAs, as they can migrate to the surface of adhesives and cause adhesion breakage. Kim et al. [20] reported on the production of acrylic water emulsions, which were later used to form PSAs from water-borne commercial dispersions with various colophony esters. Resins with a lower $T_{g}$ have practical advantages, as they are more susceptible to forming stable (easier to produce) water dispersions under normal pressure conditions [19]. Aydin et al. [21] introduced a method for obtaining water solvent-free dispersions of polymer and tackifier that can be used to produce colophony dispersions for incorporation into PSAs. Geoghegan and Wang [22] described the laboratory production of resin-in-water dispersions using a resinic ester with a $T_{g}$ of 83 ${}^{\circ }C$. Boonstra et al. [23] used a 4 L reactor with temperature control to prepare resin-in-water dispersions, which were further blended with acrylic latex to produce PSAs. Miller [24] proposed a formulation where a resinic ester combined with an antioxidant—butylated hydroxytoluene—was utilized to produce the dispersion. Aarts et al. [25] produced dispersions of tackifiers using glycerol ester. Finally, Yang et al. [26] reported on the production of resin dispersions with tackifier in continuous mode. This rich body of knowledge, which is available in the form of patents, was crucial in setting up the experimental installation and emulsion characterization during this study.

The design and development of products is a well-established field and industry is constantly adopting new practices to accelerate and systematize procedures. Companies recognize the potential of these practices to (i) respond successfully to market uncertainty and speed; (ii) improve knowledge and systematize creative processes; and (iii) ensure that decision-making processes are explicit and well-documented [27]. Despite the increasing importance of this field in academia and industry, the literature on the systematic development of new water-based emulsions for PSAs remains scarce, especially because the process needs to be compatible with customers’ specifications and the extensive experimental work required.

This paper aims to fill this gap. Here, we propose a sequential approach to the systematization of procedures. The proposed strategy is aligned with current practices, specifically, those that integrate the needs of the customer into the various stages of concept (and product) design [28]. Fundamentally, it relies on a combination of experimental work planned to maximize the obtained information and proper customer-centric decision-making tools to identify the optimal candidate concepts (in the form of emulsions or designated formulations) to develop. To identify good candidate emulsions and optimize their performance, we use the design of experiments. To systematically compare the metrics representing the product quality interpreted by customers, we use (i) engineering methods for robust design [29], specifically the concept of the loss function introduced by Taguchi et al. [30]; and (ii) multicriteria decision-making methods to aggregate multiple quality metrics into a single performance indicator.

This paper presents the following three novel elements: (i) a systematic customer-centered approach for developing polymer-based products; (ii) the use of robustness-centered techniques for assessing the performance of candidate emulsions, along with multi-attribute value techniques for integrating individual quality criteria into a performance metric; and (iii) a methodology based on the sequential design of experiments for optimizing formulations. The proposed approach is illustrated through the development of resin-in-water emulsion systems for use in PSA production.

In the remaining sections, boldface lowercase letters represent vectors, boldface capital letters represent continuous domains, blackboard bold capital letters represent discrete domains, and capital letters represent matrices. Finite sets containing ι elements are compactly represented by 〚ι〛≡{1,⋯,ι}. The transpose operation of a matrix/vector is represented by “${}^{⊺}$”.

This paper is organized as follows. Section 2 presents (i) the materials used in the experiments; (ii) the experimental equipment; (iii) the laboratory procedure; and (iv) the equipment used for quality characterization. Section 3 systematizes the product development procedure used herein and briefly outlines (i) the *loss function* used to select the best products by considering customers’ specifications, and (ii) the linear model used to represent the multi-criteria decision-making tool utilized for ranking. Section 4 applies the proposed approach to the design of physically viable resin-in-water emulsions. First, we select a set of promising candidate formulations. Then, we optimize them using a sequential optimal design of experiments, where the goal is to maximize the global performance metric by combining four quality characteristics. Finally, we analyze the repeatability of the production method. Section 5 provides an overview of this work and a summary of the results.

## 2. Materials and Equipment Characterization

This section introduces the experimental procedure. In Section 2.1, we characterize the materials used. In Section 2.2, the equipment used for the experimental work is presented and the procedure is described. Finally, Section 2.3 describes the measurement equipment used for characterizing the emulsions.

The resin industry commonly uses the softening point, $T_{\mathrm{soft}}$ as a measure of the glass transition temperature rather than differential scan calorimetry, as it corresponds to the point at which a specific resin probe slides into polymer due to a temperature increase. A physical test is used to determine the softening temperature, where a standard probe of specific dimensions and weight is gradually heated until it starts flowing. Typically, the glass transition temperature of colophony resins is about 30–40 ${}^{{}^{\circ}}C$ below the softening point, which, in turn, is highly correlated with the average molecular weight. Higher softening temperatures correspond to resins with higher average molecular weights. In our study, we used the ASTM D1525-17 standard for measuring the softening temperature [31].

### 2.1. Materials

The resins used in the experiments were modulated regarding their softening point. That is, we used two resins processed under different conditions, R1 and R2, to combine new products with a given $T_{\mathrm{soft}}$. This part of the experimental work involved various (sequential) tests, where the compositions of the mixtures in R1 and R2 were varied until the new resins obtained (denoted here as A, B, and C (see Table 1)) satisfied the target values of $T_{\mathrm{soft}}$ shown in the right column in the table. Essentially, A, B, and C are the *mother* resins used to produce the resin-in-water emulsions (see Section 4). Here, our goal was to regulate the impact of resin quality, as quantified by the softening temperature, on emulsions.

We used three surfactants in the experimental work. Due to industrial property limitations imposed by the suppliers, we cannot name them. The choice of anionic surfactants for use in the emulsions was based on both prior industrial knowledge and the need to ensure compatibility with the resin systems under consideration. Here, they are denoted as S1 (surfactant of supplier 1), S2 (surfactant of supplier 2), and S3 (surfactant of supplier 3). All other reagents used, such as potassium hydroxide (KOH) and water, were of analytical grade.

### 2.2. Experimental Equipment and Procedure

The experimental work was conducted using a purpose-built system consisting of a jacketed reactor vessel of 500 mL with a batch-mode agitator. The temperature along the batches was automatically controlled and the agitator was designed to ensure ideal mixture conditions, with its geometry preventing the accumulation of solids and its ability to generate a vortex capable of efficiently dispersing the water phase when added to the system.

Each experiment required the configuration of five parameters that could potentially affect the quality of the resin-in-water emulsions, namely the softening temperature of the resin, the surfactant, the amount of KOH added to the system during the reaction (expressed in %wt relative to the resin mass), the agitation rate of the agitator (measured in rpm), and the amount of surfactant required (also expressed in %wt relative to the resin mass).

After selecting the experimental conditions, 200 g of resin was fed into the reactor (see Figure 1a). Then, the temperature control module and heating system were switched on and the temperature at which the emulsion is prepared was set to correspond to the softening point of the resin. The agitator was also switched on and adjusted to the previously determined speed. After two hours when the resin reached a stable and nearly constant temperature and became fluid (see Figure 1b), a given volume of a solution with a 1:1 mass ratio of KOH was added to the reactor. The resulting solution was then agitated for 30 min, which corresponds to the time period during which the saponification reaction occurs. Next, the surfactant was added in small amounts. The resulting solution, now containing the surfactant, was agitated for another 30 min to ensure homogeneity. Finally, the water was added in increments in all the experiments. Essentially, the same relative amount of water was added at equal time points to minimize the effects of different mixing strategies in the emulsion characteristics. The amount of water added was calculated such that at the end of the process, the amount of solids in the water phase was about 55 %wt. Next, we describe the physical/chemical transformations that occurred during the formation of the resin-in-water emulsions.

Initially, a water-in-resin emulsion was formed. During this time, a small amount of water (in small increments) was added to the system. This dispersion was slowly stirred to yield a concentrated resin-in-water emulsion. This period was characterized by phase inversion as a result of sharp changes in viscosity and electrical conductivity. The emulsion formed was then diluted by adding more water in larger increments. The phase inversion was visually easy to identify and characterizes the point at which the dispersion water-in-resin phase forms the resin-in-water emulsion (see Figure 1c).

### 2.3. Quality Characterization Equipment

To characterize the quality characteristics of the emulsions we considered four variables: (i) viscosity (expressed in cP); (ii) pH (non-dimensionally expressed); (iii) solid content (expressed in %wt); and (iv) size of the particles (represented here by the average diameter, expressed in nm). Viscosity was measured with a *Brookfield DV2T spindle RV03, Brookfield, Toronto, Canada* viscosimeter. pH was measured using a *Mettler Toledo SevenEasy^TM^ pH meter, Mettler Toledo, Greifensee, Switzerland*. The solid content was measured using gravimetric analysis with a precision balance. The particle size was measured using a *Malvern Zetasizer Nano ZS, Malvern, Worcestershire, United Kingdom* system, which provided both the particle size distribution (PSD) and the cumulative distribution curves.

The stability of an emulsion depends on various parameters, with one of the most significant being the size and distribution of the particles—the smaller the dispersed particles, the stabler the system. More specifically, according to industry research on this type of emulsion, when the particle size is lower than 500 nm and has a unimodal distribution, good results in terms of stability are typically achieved. Thus, by controlling the particles’ size and the unimodality of the respective PSD, as in our experiments, the stability of the formulations, or at least the identification of formulations that may not be stable, can be indirectly ensured. We also note that during the optimization stages, visual evaluation of the emulsions was performed and some of them were reported as non-dispersed (ns), as shown in the tables of results (see Section 4.2, Section 4.3 and Section 4.4). Moreover, the initially produced samples were observed after three months and no significant phase separation was observed, which is an excellent indicator of stability and demonstrates the compatibility of the resin system, including the resin–surfactant combination and the production process.

## 3. Development Approach and Related Tools

In this section, we present an overview of the methods used to select the formulations that were worth exploring further and the procedure used to design resin-in-water emulsions tailored to customers’ needs. Section 3.1 explains the methodology used to create a performance metric that captures customer satisfaction relative to all the quality characteristics (see Section 2.2). Section 3.2 discusses the procedure used to design the product.

### 3.1. Overall Quality Performance Metric

Here, we describe the construction of an overall performance metric that could represent the quality of the emulsions with a customer-centric approach [32]. Two separate challenges emerged during this process: (i) how to measure the quality of the formulations according to each quality characteristic specification; and (ii) how to aggregate the metrics representing each quality characteristic into an overall performance index.

To measure the quality of the emulsions, we assumed the customers’ specifications were known and we adopted the Taguchi method, specifically the loss function. The formulation of the loss function depends on the quality characteristic under consideration, that is, *smaller-is-better*, *larger-is-better*, or *target-is-best* [33]. These quality characteristics represent the loss in economic value associated with missing the target (i.e., the quality goal that fully satisfies customers). This approach is commonly used to address practical engineering problems (see Liao and Kao [34] for an example).

Let the quality characteristics be designated by $C_{i}$, where *i* is the number of quality characteristics of interest, i∈〚q〛, and *q* is the number of quality dimensions used to characterize the product. In our case, q=4, i=1 represents the viscosity, i=2 represents the pH, i=3 represents the solid content, and i=4 represents the average particle diameter. Let C be the vector containing all the quality characteristics. The loss function for the quality characteristics $C_{i}$, which is represented by $L(C_{i})$, is:(1)$$L\left(C_{i}\right)=\left\{\begin{array}{cc} k_{i}\;{\left(C_{i}-\xi_{i}\right)}^{2} & \mathrm{for}\;\mathrm{scenarios}\;\mathrm{target-is-best}\\ k_{i}\;C_{i}^{2} & \mathrm{for}\;\mathrm{scenarios}\;\mathrm{smaller-is-better}\\ k_{i}/C_{i}^{2} & \mathrm{for}\;\mathrm{scenarios}\;\mathrm{larger-is-better} \end{array}\right.,$$
where $k_{i}$ is the loss constant for the *i*th quality characteristic and $\xi_{i}$ is the respective target value. We note that $L(C_{i}):\;R\to R,\;i\in 〚q〛$. Consequently, Equation (Equation 1) does not allow operating (e.g., summing or subtracting) different loss functions since they can have different domains [35]. To assure their commensurability, we normalize them using the respective specifications and target values. The normalized loss function used for measuring quality, therefore, is
(2)$$L^{\mathrm{norm}}\left(C_{i}\right)=\left\{\begin{array}{cc} 4\;{\left(C_{i}-\xi_{i}\right)}^{2}/{\left(US_{i}-LS_{i}\right)}^{2} & \mathrm{for}\;\mathrm{scenarios}\;\mathrm{target-is-best}\\ C_{i}^{2}/US_{i}^{2} & \mathrm{for}\;\mathrm{scenarios}\;\mathrm{smaller-is-better}\\ LS_{i}^{2}/C_{i}^{2} & \mathrm{for}\;\mathrm{scenarios}\;\mathrm{larger-is-better} \end{array}\right.,$$
where $LS_{i}$ and $US_{i}$ are the lower and upper specifications for the *i*th quality characteristic, respectively. Consequently, $L^{\mathrm{norm}}\left(C_{i}\right):\;R\to \left[0,1\right],\;i\in 〚q〛$.

The aggregation of multiple criteria for measuring the performance relative to the quality characteristics into a performance metric was based on multi-attribute value theory (MAVT) (see Belton and Stewart [36]). MAVT is a multicriteria decision method that is used to address situations where all (finite) alternatives are known with complete certainty, and the objective function aggregates the value scores that reflect each alternative’s performance on each criterion, appropriately weighted. Typically, the weights reflect the relative importance that the decision maker gives to each criterion. Various model forms can be used to represent the aggregation but the most common in practical applications is the linear model, as it simplifies the elicitation of the weights [37]. It should be noted that the value score functions must be commensurable but may represent conflicting criteria. Examples of the application of MAVT to practical decision problems can be found in the literature (see, for example, Keeney [38]. The elicitation of the weights is conducted via surveys [39] or based on expert-based knowledge [40].

The overall quality metric used herein is, therefore,
(3)$$O=\sum_{i=1}^{q}w_{i}\;L^{\mathrm{norm}}\left(C_{i}\right),$$
where the *q*-element column vector w contains the weights of all single-score criteria and they obey the constraints $\sum_{i=1}^{q}w_{i}=1,\;w_{i}\geq 0,\;i\in 〚q〛$. The promising emulsions minimize (Equation 3) with respect to $C_{i},\;i\in 〚q〛$.

### 3.2. Design Procedure

Now, we introduce the product development procedure used in this study, which involved multiple stages of experimental work. The main goal was to identify a set of promising formulations that could later be tested for their potential to scale up. Very often the number of quality characteristics that impact product quality perception is large, as is the number of factors that impact their performance. As a result, the experiments for factor screening have to be minimized, as they require a large amount of work and resources. Therefore, an approach based on splitting the experimental work into three sequential phases where some factors are fixed in some of them is beneficial. This approach minimizes the resources required while allowing for the analysis of a large range of concepts and preventing the discarding of promising formulations. Our approach involved dividing the experimental work into three stages: the first stage involved factor/quality criteria screening; the second stage involved optimization; and the third stage involved the study of the repeatability features of the formulations considered most promising. Figure 2 illustrates this sequence of stages and shows that the procedure analyzed here was included in the concept development phase, where the goal was to choose a small set of candidate formulations to later test regarding their potential to scale up. The concept development was carried out after performing a systematic analysis of the product’s most relevant quality characteristics, the specifications that must be met, and the factors that typically affect technical performance, as well as the company’s strategic planning regarding product development. Each stage of the experimental design had a specific goal and was followed by experimentation and an analysis of the results. In each of the stages of the concept design phase, the decision tool used for ranking (and optimizing) the formulations was the overall quality metric (Equation 3) that focuses on the customer targets. It must be emphasized that in the first stage of the procedure, some of the experimental factors were fixed to reduce the number of experiments for screening purposes. Then, the formulations identified in the screening were optimized, i.e., the factors initially screened were fixed, but another group was varied.

Here, we analyze each of the stages. Stage 1 involved screening a subset of factors judged as the most relevant for the quality metrics. The complete set of factors was divided into (i) primary factors, $x^{\mathrm{pri}}$, and (ii) secondary factors, $x^{\sec}$, where the former were varied in the screening experiments and the latter were fixed to an appropriately normal level. Thus, $x\equiv x^{\mathrm{pri}}\cup x^{\sec}$ represents the complete set of factors populated with *k* elements. The factors were divided into experience-based knowledge from a company that produces colophony-resin derivatives and the requirements of industrial customers. This ranking also captured the ease of modifying the factors at the experimental level and their impact on the overall performance of the product. The primary factors were modified to produce a set of feasible emulsions and the secondary factors were used to optimize their performance.

Among the primary factors, there were continuous and discrete (categorical) inputs, and we assumed that their discrete levels could be ordered [41]. In this phase, we adopted a full (or partial) factorial design of experiments after deciding which levels were to be used for the experiments and the corresponding codes. Details about the choice of the experimental designs are discussed in state-of-the-art references such as Montgomery [42]. Computational tools such as JMP^®^ can be used for the design and data analysis [43]. The results of Stage 1 included a set of primary factors and levels that could optimize quality performance and this combination of factors and levels was fixed in the second stage.

In Stage 2, we adopted a different approach to optimizing the quality performance. Here, we varied the factors belonging to the secondary group *one at a time* using a sequential optimal design strategy. We assumed that they were continuous and the goal was to minimize (Equation 3) with respect to $x_{i}^{\sec}\in x^{\sec}$, where *i* is the index for the elements of the vector containing the secondary factor indices, $〚k^{\sec}〛$, where $〚k^{\sec}〛\subset 〚k〛$. The sequential optimal design of experiments is a commonly used technique in practical applications (see, e.g., Box and Hunter [44], Goujot et al. [45]).

The sequential optimal design of experiments is an approach consisting of two distinct phases. In the first phase, a set of experiments is conducted to fit a causal model between the factors and the response variable(s). In the second phase, the model is used to determine the factor levels for the next experiment to maximize the information criterion. The second phase is then iterated and in each iteration, the causal model is updated using the observations of the latest experiment and a new experiment is prescribed. Here, we adapted a strategy from the general framework, where the regression models relating the secondary factors to the quality variables were linear or square polynomial functions. Instead of maximizing an information criterion, we minimized the overall quality metric (Equation 3) in each iteration.

Now, we describe the complete procedure. First, we conducted experiments where the level of a secondary factor, say the *j*th factor ($x_{j}\in x^{\sec}$), was varied. We used the results of the observations from Stage 1 to fit a linear model that related each of the quality characteristics with the factor, i.e.,
(4)$$C_{i}=g_{1,i}\left(x_{j}\right)$$
where $g_{1,i}\left(•\right),\;i\in 〚q〛$ are linear functions that depend on $x_{j}$. Next, the obtained regression models $g_{1,i}\left(•\right)$ were used to prescribe the next experiment by solving the optimization problem
(5a)$$\begin{array}{cc} \min_{x} & \;\sum_{i=1}^{q}w_{i}\;L^{\mathrm{norm}}\left(C_{i}\right) \end{array}$$
(5b)$$\begin{array}{cc} s.t. & \;C_{i}=g_{1,i}\left(x_{j}\right),\;i\in 〚q〛. \end{array}$$

Problem (5) belonged to the nonlinear programming class and was solved with appropriate tools. Equation ([Disp-formula FD5a-polymers-15-01691]) served as the objective function (i.e., the overall quality metric), and ([Disp-formula FD5b-polymers-15-01691]) represented a set of equality constraints that reflected the causal relations described by the previously fitted regression models.

The next experiment provided one additional observation, which was used to update the models $g_{1,i}\left(•\right)$. In practice, when we have the results from more than two experiments, we replace the linear regression models with square polynomial models, i.e., (Equation 4) was replaced by
(6)$$C_{i}=g_{2,i}\left(x_{j}\right),$$
and after model fitting, this new equality relation was used in the next optimization step instead of ([Disp-formula FD5b-polymers-15-01691]). The model fitting problems were linear with respect to the parameters and were solved using the least squares algorithm [46].

The accumulation of experiments allowed us to converge to the goal since the procedure had convergence properties (see [47]). However, to rationalize the experimental work, we set a given tolerance for the distance between the quality performance and the optimum (O=0.0), and the cycle ended when it was attained. In this case, we set the tolerance to 0.10. The complete optimization may require successive sequential experimental procedures for different factors, i.e., one may apply the procedure by varying the secondary factor $x_{j}$ in Equations (Equation 4) or (Equation 6).

In Stage 3, we analyzed the repeatability of the promising formulations. The factors were fixed to the optimal levels obtained in previous stages and new experiments were conducted by replicating the conditions. The complete set of observations obtained for the optimum formulations was used to determine the average value, ${\bar{x}}_{C_{i}}$, the standard deviation, $s_{C_{i}}$, and the coefficient of variation (expressed in %) for each quality characteristic $C_{i}$. Here, we considered
(7)$$C_{v,i}=\frac{s_{C_{i}}}{{\bar{x}}_{C_{i}}}\times 100\%\;$$
where $C_{v,i},\;i\in 〚q〛$ is the coefficient of variation for the *i*th quality characteristic. To assure the repeatability of the experimental results (and the procedure), the coefficient of variation should ideally be small.

## 4. Results

In this section, we utilized the approach introduced in Section 2 and Section 3 to the design of resin-in-water emulsions for application in PSA fabrication. In Section 4.1, we characterize the design problem and the decision-making tool used to assess the quality performance of the formulations. In Section 4.2, we apply the screening experimental design discussed in Section 3.2 to find the optimal levels of the primary factors. In Section 4.3, we used sequential experimental design to optimize the formulations by judiciously choosing the levels of the secondary factors. Finally, in Section 4.4, we analyze the experimental repeatability of the formulations.

### 4.1. The Design Problem

Here, we characterize the design context and introduce the overall quality metric used to assess the formulations resulting from the experiments.

The quality characteristics were introduced in Section 2.3, where $C_{1}$ represented viscosity, $C_{2}$ represented pH, $C_{3}$ represented the solid content, and $C_{4}$ represented the particle diameter, and they were measured using the methods described in Section 2.3. The factors affecting the quality characteristics were listed in Section 2.1 and Section 2.2. The primary factors were the resin, which was characterized by the respective softening temperature and mathematically represented by $x_{1}$, and the surfactant, $x_{2}$. Both factors were discrete. The secondary factors were the amount of KOH added to the reactional system, represented by $x_{3}$; the agitation rate, $x_{4}$; and the amount of surfactant, $x_{5}$. All secondary factors were continuous and were used to optimize the formulations regarding the quality characteristics.

The product specifications gathered from the customers are listed in Table 2. Normalized loss functions were constructed for all the quality characteristics by applying Equation (Equation 2), and they are shown in the second column in Table 3.

The weights representing the relative importance of the quality characteristics were determined using expert-based knowledge and typical customer requirements. They are shown in the third column in Table 3. Consequently, the overall quality metric, which was obtained using Equation (Equation 3) and used for ranking the emulsions in the remaining sections, is
(8)$$O=3\times 10^{-7}\;C_{1}^{2}+0.15\;{\left(C_{2}-8\right)}^{2}+0.05\;{\left(C_{3}-55\right)}^{2}+5\times 10^{-7}\;C_{4}^{2}.$$

Besides this metric, an additional feature emerged during the process of extraction of experience-based knowledge. The technical staff and customers recommended that for subsequent applications, the particle size distribution (PSD) should be unimodal, as it is a common indicator of the stability of the resulting emulsion. To account for this requirement, a new binary quality characteristic was introduced into the analysis, $C_{5}$, where
(9)$$C_{5}=\left\{\begin{array}{cc} 0 & \mathrm{for}\;\mathrm{unimodal}\;\mathrm{PSD}\\ 1 & \mathrm{for}\;\mathrm{non}\text{-}\mathrm{unimodal}\;\mathrm{PSD}. \end{array}\right.$$

This characteristic was not explicitly included in the decision model (Equation 8) but non-unimodal PSD emulsions (i.e., $C_{5}=1$) were automatically discarded from the group of promising formulations.

### 4.2. Screening of Primary Factors

This section presents the work conducted in Stage 1 of the proposed approach (see Figure 2), which was used for screening the primary factors.

The vector of factors included five elements, i.e., $x={\left(x_{1},x_{2},x_{3},x_{4},x_{5}\right)}^{⊺}$, where $x_{1}$ represented the resin composition, distinguished by the respective softening temperature; $x_{2}$ represented the surfactant, identified by the supplier; $x_{3}$ represented the amount of KOH used; $x_{4}$ represented the agitation rate; and $x_{5}$ represented the amount of surfactant. Consequently, the number of factors of the design problem, *k*, was 5. $x_{1}$ and $x_{2}$ represented the primary factors, that is, $x^{\mathrm{pri}}={\left(x_{1},x_{2}\right)}^{⊺}$ and $k^{\mathrm{pri}}=2$. $x_{3}$, $x_{4}$, and $x_{5}$ represented the secondary factors, that is, $x^{\sec}={\left(x_{3},x_{4},x_{5}\right)}^{⊺}$ and $k^{\sec}=3$.

For the screening of the primary factors, we used a three-level full factorial design of experiments (see Section 3.2). Specifically, the complete design consisted of $3^{k^{\mathrm{pri}}}=9$ experiments, where $x_{1}$ and $x_{2}$ were varied in the three levels. Table 4 shows the levels tested for each factor and the respective codes and designations (shorten to *Designat.*) used in the remaining parts of the paper. In all the experiments carried out in this stage, the secondary factors were kept constant: the amount of KOH added was 2 %wt relative to the resin, the amount of surfactant was 7 %wt relative to the resin, and the agitation rate was set to 100 rpm. Furthermore, the experimental temperature was maintained at 25 ${}^{\circ }C$ in all the tests. The water was added gradually to obtain dispersions with a solid content within the range of 55±1 %wt.

Table 5 presents the results of the experiments. The first column shows the experiment number, the second column shows the $x_{1}$ level, the third column shows the $x_{2}$ level, and the fourth column shows the designation of the emulsion, where the first two symbols denote the base resin used (see Table 4), the fourth and fifth symbols denote the surfactant, and the last digit indicates the experiment number under the same conditions (i.e., for a given pair, $x_{1}$ and $x_{2}$). Furthermore, columns 5 to 8 show the results of the characterization of the emulsions, and column 9 shows the average diameter, where 95% of the particles had a smaller size, $d_{95}$, which, in turn, is an extreme measure of the PSD obtained through image analysis (see Section 2.3). Finally, column 10 shows the information on the unimodality of the PSD curve, and column 11 shows the overall performance metric (Equation 8). It is important to note that, similar to the average particle diameter, $d_{95}$ should be as small as possible.

The analysis of the results showed that (i) three of the formulations did not produce stable emulsions (i.e., formulations A2:S3, A3:S1, and A3:S3), which may indicate that resins with lower $T_{\mathrm{soft}}$ (lower average molecular weight) have better emulsification properties in water; (ii) the PSD curves obtained for three of the formulations were not unimodal (i.e., formulations A1:S3, A2:S1, and A3:S2), which may indicate that they were physically unstable); (iii) the most promising formulation obtained from the resin with $T_{\mathrm{soft}}=60\;{}^{\circ }C$ was A1:S2, which was optimized in Stage 2; and (iv) the most promising emulsion obtained from resin A2 was A2:S2, which was also optimized in Stage 2.

To avoid an overly strict limitation of the candidate alternatives, which is common in product design, formulation A1:S1 was also considered for optimization despite its sub-optimal performance (see the value *O* in the first line in Table 5). This was because (i) the concepts resulting from the subsequent stages were widened if at least two surfactants were included in the set of formulations to optimize, and (ii) the deficiency of formulation A1:S1 was related to a single quality characteristic—the viscosity (see the value $C_{1}$ in the first line in Table 5)— indicating good performance in all other criteria. In summary, formulations A1:S1, A1:S2, and A2:S2 were optimized, which is discussed in Section 4.3. $x_{1}$ and $x_{2}$ were fixed to −1 or 0 and none of the formulations that were optimized were based on the resin with a higher $T_{\mathrm{soft}}$.

### 4.3. Optimization of Secondary Factors

Now, we optimize the promising formulations identified in Stage 1. The secondary factors used in the optimization were the amount of KOH used relative to the resin weight, $x_{3}$; the agitation rate, $x_{4}$; and the amount of surfactant relative to the resin weight, $x_{5}$. The sequence of the experiments followed the procedure described in Section 2.2. In contrast to Stage 1, here, we did not use the previously identified factorial plan but followed the sequential optimal design-based approach introduced in Section 3.2 for each emulsion. The results are shown in Table 6, which can be interpreted similarly to Table 5.

The accumulation of experiments followed a sequential order, where the next experiment was prescribed based on the updated model(s) relating the factors to the quality characteristics. It should be noted that the optimization procedure required a different number of experiments for each emulsion (seven experiments for A1:S1 and four for the other two formulations). This was due to the fact that the models fitted with the previously obtained observations had a lower prediction ability, as they relied on a small and sometimes dispersed sample of points. The experiments from Stage 1 were considered the reference conditions and were incorporated into the sequence as the initial condition. The convergence was achieved with an error of 0.053 for A1:S1 and less than 0.03 for the other two formulations, values considered within the specified tolerance. The trajectory to the optimum passed through conditions where no emulsion was formed (see Experiment 11) or the PSD was not unimodal (see Experiment 14), both indicating unstable formulations.

Table 7 presents a summary of the most promising formulations identified in this phase, which are analyzed regarding their experimental repeatability in Section 4.4. It is important to note that the optimal conditions for emulsions A2:S1 and A2:S2 were identical, so the final decision about the most promising formulation requires an accurate economical analysis.

### 4.4. Formulations’ Repeatability

Here, we assess the repeatability of the experiments conducted to produce the optimal formulations shown in Table 7. In our context, repeatability refers to the ability to produce formulations with similar performance quality characteristics under the same experimental conditions. This measure is crucial in the industrialization of a concept, as it represents the inherent variation in the production process (i.e., precision). Processes with lower precision may require more accurate control systems to ensure that the customers’ specifications are systematically achieved. The repeatability analysis followed the guidelines described in Section 3.2 and the coefficient of variation was determined using Equation (Equation 7).

We replicated the optimal formulations by conducting three additional experiments for each formulation. The optimization experiments from Stage 2 (see Table 7) were also included in the data set used for analysis. The complete set of observations is shown in Table 8. The results showed that the coefficient of variation was below 10% for all the quality characteristics, with the exception of viscosity and the particles’ $d_{95}$ for A1:S1. Additional experiments for this formulation may be necessary to further assess its repeatability. Overall, the results demonstrated the repeatability of the production procedure and the resin–surfactant combinations.

## 5. Conclusions

We consider the problem of designing high-added value products from natural compounds and propose a systematic approach that uses customer specifications to “drive” the experimental work in the concept development phase. Our procedure consists of three main steps: (i) screening the primary factors using a full factorial experimental design; (ii) optimizing the formulation using a sequential experimental design, where the secondary factors are manipulated to iteratively improve the quality performance; and (iii) assessing the experimental repeatability of the optimal formulations. The complete procedure is described in detail in Section 3.2. The tool used to rank the experimentally obtained emulsions aggregates the loss functions representing the customer quality perception with respect to the single quality characteristics. Multi-attribute value theory provides the theoretical framework for the aggregation, and the relative weights representing the importance of each quality criterion are elicited from the preferences of experts and customers. The concept of the loss function introduced by Taguchi is employed to represent the quality performance relative to each individual characteristic.

The approach proposed herein is successfully used to develop pine-resin-in-water emulsions for the fabrication of pressure-sensitive adhesives (see Section 4). The next stage of product development requires the scale-up of the optimal formulations to anticipate technological issues and construct a more accurate economic analysis that, in turn, will enable a more robust decision on its industrialization. We believe the proposed tool can be applied to other products, especially in cases where the design procedure requires extensive experimental work, the time to market is short, and the resources are limited.

## Figures and Tables

**Figure 1 polymers-15-01691-f001:**
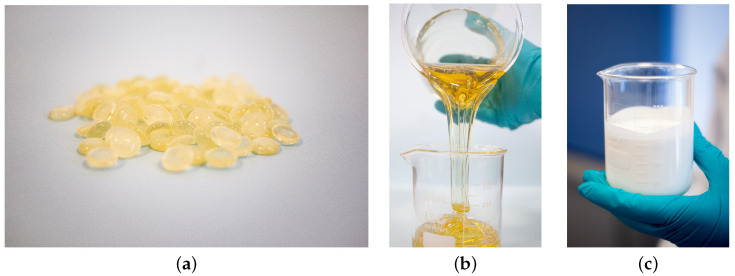
The state of the resin during the process (**a**) as a (solid) raw material; (**b**) after heating to the softening point; (**c**) as a resin-in-water emulsion.

**Figure 2 polymers-15-01691-f002:**
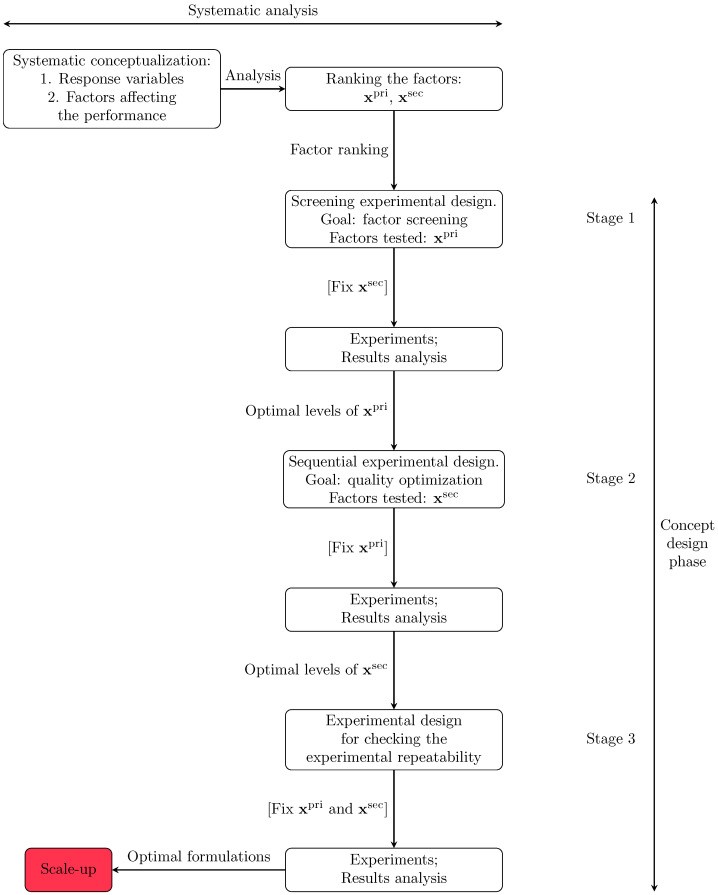
Systematic procedure for hierarchically organizing the experimental work in a customer-centric approach.

**Table 1 polymers-15-01691-t001:** Characterization of the resins used in the experimental work.

Resin Designation	$T_{\mathrm{soft}}$ (${}^{\circ }C$)
A	60
B	70
C	80

**Table 2 polymers-15-01691-t002:** Specifications imposed on the quality characteristics of the resin–water emulsions.

Quality Characteristic	Lower Specification	Upper Specification	Target Value	Loss Function
Viscosity (@ 25 ${}^{\circ }C$)	ns${}^{\;†}$	1000 cP	ns${}^{\;†}$	*smaller-is-better*
pH	7	9	8	*target-is-best*
Solid content	54 %wt	56 %wt	55 %wt	*target-is-best*
Particle diameter	ns${}^{\;†}$	1000 nm${}^{\;‡}$	ns${}^{\;†}$	*smaller-is-better*

^†^ undefined; ^‡^ 95% of particles below 1000 nm.

**Table 3 polymers-15-01691-t003:** Normalized loss functions and relative weights.

Quality Characteristic	Normalized Loss Function	Weight ($w_{i}$)
Viscosity (@ 25 ${}^{\circ }C$)	$L^{\mathrm{norm}}\left(C_{1}\right)=1\times 10^{-6}\;C_{1}^{2}$	0.30
pH	$L^{\mathrm{norm}}\left(C_{2}\right)={\left(C_{2}-8\right)}^{2}$	0.15
Solid content	$L^{\mathrm{norm}}\left(C_{3}\right)={\left(C_{3}-55\right)}^{2}$	0.05
Particle diameter	$L^{\mathrm{norm}}\left(C_{4}\right)=1\times 10^{-6}\;C_{4}^{2}$	0.50

**Table 4 polymers-15-01691-t004:** Levels and codes of primary factors.

Factor	Level	Characterization	Designat.
	−1	Resin with $T_{\mathrm{soft}}=60\;{}^{\circ }C$	A1
$x_{1}$	0	Resin with $T_{\mathrm{soft}}=70\;{}^{\circ }C$	A2
	+1	Resin with $T_{\mathrm{soft}}=80\;{}^{\circ }C$	A3
	−1	Surfactant from S1	S1
$x_{2}$	0	Surfactant from S2	S2
	+1	Surfactant from S3	S3

**Table 5 polymers-15-01691-t005:** Results of the experimental design in the *screening of primary factors* stage.

# Exper.	$x_{1}$	$x_{2}$	Designat.	$C_{1}$ (cP)	$C_{2}$ (-)	$C_{3}$ (%wt)	$C_{4}$ (nm)	$d_{95}$ (nm)	$C_{5}$	*O*
1	−1	−1	A1:S1_1	1562	11.04	54.36	209.6	310.45	0	2.161
2	−1	0	A1:S2_1	265	8.63	55.11	428.2	1029.93	0	0.173
3	−1	+1	A1:S3_1	1292	11.19	54.58	614.9	1302.51	1	2.225
4	0	−1	A2:S1_1	187	10.28	55.62	444.3	4246.54	1	0.908
5	0	0	A2:S2_1	339	8.46	54.34	262.7	695.22	0	0.123
6	0	+1	A2:S3_1	-	-	-	-	-	-	ns${}^{\;†}$
7	+1	−1	A3:S1_1	-	-	-	-	-	-	ns${}^{\;†}$
8	+1	0	A3:S2_1	112	8.55	54.44	615.5	1064	1	0.254
9	+1	+1	A3:S3_1	-	-	-	-	-	-	ns${}^{\;†}$

^†^ non-dispersed formulations.

**Table 6 polymers-15-01691-t006:** Results of the experimental design in the *optimization of secondary factors* stage.

Optimization of Emulsion A1:S1 ($x_{1}=-1$, $x_{2}=-1$)										
**# Exper.**	**$x_{3}$ (%wt)**	**$x_{4}$ (rpm)**	**$x_{5}$ (%wt)**	**Designat.**	**$C_{1}$ (cP)**	**$C_{2}$ (-)**	**$C_{3}$ (%wt)**	**$C_{4}$ (nm)**	** $C_{5}$ **	** O **
1	2.0	100	7.0	A1:S1_1	1562	11.04	54.36	209.6	0	2.161
10	2.5	100	7.0	A1:S1_2	5967	10.11	55.02	150.0	0	11.361
11	0.0	100	7.0	A1:S1_3	-	-	-	-	-	ns${}^{\;†}$
12	1.0	100	7.0	A1:S1_4	440	10.19	54.44	261.1	0	0.827
13	1.0	100	6.0	A1:S1_5	429	10.14	55.92	319.0	0	0.835
14	0.5	100	7.0	A1:S1_6	-	-	-	-	-	ns${}^{\;†}$
15	0.5	100	10.5	A1:S1_7	235	8.99	54.68	1007.7	1	0.676
16	0.5	100	14.0	A1:S1_8	172	8.28	55.46	209.4	0	0.053
Optimization of emulsion A1:S2 ($x_{1}=-1$, $x_{2}=0$)										
2	2	100	7.0	A1:S2_1	265	8.63	55.11	428.2	0	0.173
17	2.5	100	7.0	A1:S2_2	352	8.36	54.30	236.2	0	0.109
18	2.0	50	7.0	A1:S2_3	392	8.62	55.92	231.1	0	0.173
19	1.0	50	7.0	A1:S2_4	-	-	-	-	-	ns${}^{\;†}$
20	1.5	50	7.0	A1:S2_5	131	8.05	54.91	205.53	0	0.027
Optimization of emulsion A2:S2 ($x_{1}=0$, $x_{2}=0$)										
5	2.0	100	7.0	A2:S2_1	339	8.46	54.34	262.7	0	0.123
21	2.5	100	7.0	A2:S2_2	308	8.35	54.58	341.3	0	0.114
22	2.0	50	7.0	A2:S2_3	337	8.70	54.96	350.6	0	0.169
23	1.0	50	7.0	A2:S2_4	87	6.96	54.94	245.5	0	0.195
24	1.5	50	7.0	A2:S2_5	118	7.94	54.98	210.5	0	0.027

^†^ non-dispersed formulations.

**Table 7 polymers-15-01691-t007:** Optimal formulations resulting from the *optimization of secondary factors* stage.

# Exper.	Designat.	$x_{1}$ (-)	$x_{2}$ (-)	$x_{3}$ (%wt)	$x_{4}$ (rpm)	$x_{5}$ (%wt)
16	A1:S1_8	−1	−1	0.5	100	14.0
20	A1:S2_5	−1	0	1.5	50	7.0
24	A2:S2_5	0	0	1.5	50	7.0

**Table 8 polymers-15-01691-t008:** Results of the experimental design in Stage 3.

Repeatability of Emulsion A1:S1 (x1=−1, x2=−1, x3= 0.5 %wt, x4= 100 rpm, x5= 14.0 %wt)
**# Exper.**	**Designat.**	**C1 (cP)**	**C2 (-)**	**C3 (%wt)**	**C4 (nm)**	** C5 **	** O **
16	A1:S1_8	172	8.25	55.46	209.43	0	0.051
25	A1:S1_9	236	8.61	54.27	460.97	0	0.205
26	A1:S1_10	231	8.42	54.43	448.80	0	0.159
27	A1:S1_11	237	8.39	55.07	449.25	0	0.141
x¯Ci		219.00	8.42	54.81	392.11		
sCi		31.44	0.15	0.56	121.92		
Cv,i(%)		14.36	1.76	1.01	31.09		
Repeatability of emulsion A1:S2 (x1=−1, x2=0, x3= 1.5 %wt, x4= 50 rpm, x5= 7.0 %wt)
20	A1:S2_5	131	8.05	54.91	205.53	0	0.027
28	A1:S2_6	132	7.96	55.26	206.35	0	0.030
29	A1:S2_7	132	8.04	54.84	202.35	0	0.027
30	A1:S2_8	128	8.01	54.87	196.25	0	0.025
x¯Ci		130.75	8.02	54.97	202.62		
sCi		1.89	0.04	0.20	4.58		
Cv,i(%)		1.45	0.50	0.36	2.26		
Repeatability of emulsion A2:S2 (x1=0, x2=0, x3= 1.5 %wt, x4= 50 rpm, x5= 7.0 %wt)
24	A2:S2_5	118	7.94	54.98	210.5	0	0.027
31	A2:S2_6	108	7.89	55.31	208.57	0	0.032
32	A2:S2_7	112	7.95	54.66	219.48	0	0.034
33	A2:S2_8	106	7.82	54.15	215.65	0	0.068
x¯Ci		111.00	7.90	54.78	213.55		
sCi		5.29	0.06	0.49	4.96		
Cv,i(%)		4.77	0.75	0.90	2.32

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
