# Peer review of "On the Design of Aqueous Emulsions of Colophony Resin"

_polymers, 2023, doi:10.3390/polym15071691_

Round 1

Reviewer 1 Report

The author show systematic design of aqueous emulsions of colophony resin, including primary factors screening; secondary factors optimization; the assessment of the formulations experimental repeatability. It was used to develop pine resin-in-water emulsions applied in the fabrication of pressure sensitive adhesives. This systematic design is very meaningful for polymer industry, I recommend this paper can be published in polymers after small revision. Specific comments are shown below:

(1)   Although Tsoft is useful for resin slides, Tg is more important to reflect the polymer chain movement, could you consider to measure it?

(2)   To characterize the quality characteristics of the emulsions, you consider the Viscosity, pH, solids content, particles’ size. Actually, the stability is vital for emulsion quality, I think you can also test this factor.

(3)   Multiattribute value theory is good, it can support the aggregation and the relative weights. Meanwhile, the TEM and GPC are recommended for characterization. 

Author Response

Dear Reviewer,
We are sending you the revised version of the Manuscript polymers-2270068 and our response letter attached. In this revision we have clarified and fixed the aspects identified by the reviewers in the previous version of the paper. We have tried our best to respond to each comment and suggestion made, and therefore have introduced changes, which are noted in color.
Our point-to-point response follows immediately each paragraph of the original report. Overall, we are grateful to both reviewers and the editorial team for a timely and careful review, which helped us to significantly improve the quality of the manuscript.
Sincerely Yours,
Belmiro P.M. Duarte

Reviewer 2 Report

Dear Authors,

the paper is an interesting application of Taguchi and MCDM methods.

Some comments about the paper:

1. The title is not so attractive: “..the systematic design….” is too elusive.

2. The procedure layout described in Figure 1 is not well formalized.

3. Line 272. How do you split the factors into the most relevant and not?

4. And then (lines 348-353), according to what do you have chosen the primary and secondary factors?

5. Conclusions are somewhere repetitious.

Author Response

(The authors gave the same response as above.)
